# Variable Selection and Allocation in Joint Models via Gradient Boosting Techniques

**Colin Griesbach** [1,*] **, Andreas Mayr** [2] **and Elisabeth Bergherr** [1]

1. Chair of Spatial Data Science and Statistical Learning, Georg-August-Universität Göttingen, 37073 Göttingen, Germany
2. Department of Medical Biometrics, Informatics and Epidemiology, University Hospital Bonn, 53127 Bonn, Germany
* Correspondence: colin.griesbach@uni-goettingen.de

**Abstract:** Modeling longitudinal data (e.g., biomarkers) and the risk for events separately leads to a loss of information and bias, even though the underlying processes are related to each other. Hence, the popularity of joint models for longitudinal and time-to-event-data has grown rapidly in the last few decades. However, it is quite a practical challenge to specify which part of a joint model the single covariates should be assigned to as this decision usually has to be made based on background knowledge. In this work, we combined recent developments from the field of gradient boosting for distributional regression in order to construct an allocation routine allowing researchers to automatically assign covariates to the single sub-predictors of a joint model. The procedure provides several well-known advantages of model-based statistical learning tools, as well as a fast-performing allocation mechanism for joint models, which is illustrated via empirical results from a simulation study and a biomedical application.

**Keywords:** joint modeling; time-to-event analysis; gradient boosting; statistical learning; variable selection





## 1. Introduction

Joint models for longitudinal and time-to-event data, first introduced in [1], are a powerful tool for analyzing data where event times are recorded alongside a longitudinal outcome. If the research interest lies in the association between these two outcomes, joint modeling avoids potential bias arising from separate analyses by combining two sub-models in one single modeling framework. A thorough introduction to the concept of joint models can be found in [2], and various well-established R packages are available covering frequentist [3,4] and Bayesian [5] approaches.

Like many regression models, joint models suffer from the usual drawbacks, where proper tools for variable selection are not immediately available and computation becomes more and more infeasible in higher dimensions. In addition, joint models also raise the question of which sub-model a variable should be assigned to, i.e., should a variable $x$ have a direct impact on the survival outcome $T$, or should the potential influence be modeled indirectly by an impact of $x$ on the longitudinal outcome $y$, which then might affect $T$? This choice gets exponentially more complex with an increasing amount of covariates, and usually has to be made by researchers based on background knowledge. Boosting techniques from the field of statistical learning, however, are well-known for addressing these exact issues. Originally emerging from the machine learning community as an approach to classification problems [6,7], boosting algorithms have been adapted to regression models [8] and, by now, cover a wide range of statistical models. For an introduction and overview of model-based boosting, we recommend [9,10].

The formulation of boosting routines for joint models is, to date, still a little-developed field. The foundations were made by [11], where generalized additive models for location scale and shape (GAMLSS) were fitted using boosting techniques. Due to the multiple predictors for each single distributional parameter, these models consist of a similar structure to joint models and thus the boosting concept for GAMLSS could be adapted to joint models by [12]. Furthermore, in [13], joint models were estimated using likelihood-based boosting techniques, and [14,15] compare boosting routines for joint models with various other estimation approaches.

In the last few years, several additional developments have been made in order to enable a variable selection for joint models, usually by applying different shrinkage techniques. In [16], an adaptive LASSO estimator was constructed that estimates $L1$-penalized likelihoods in a two-stage fashion. This approach was later extended to multivariate longitudinal outcomes in [17] and to time-varying coefficients in [18]. In [19,20], Bayesian shrinkage estimators were applied to achieve either variable or model selection for various classes of joint models and, recently, ref. [21] applied Monte Carlo methods to enable a variable selection for joint models with an interval-censored survival outcome. However, all of these mentioned approaches are only capable of selecting and estimating effects into predefined predictor functions. To the best of our knowledge, no methods exist that allocate single features to the given sub-models in a data-driven way.

The aim of the present work is to combine recent developments from the field of model-based gradient boosting in order to develop a new routine, `JMalct`, that is able to allocate the single candidate variables to the specific sub-models. Therefore, the initial boosting approach by [12] was equipped with a non-cyclical updating scheme proposed by [22] and adaptive step-lengths as investigated in [23]. These two preliminary works are of high importance and their combination is the foundation of our proposed allocation procedure. Furthermore, the `JMalct` algorithm makes use of a recent random effects correction [24] providing an unbiased estimation of the random effects using gradient boosting and tuning based on probing [25] for faster computation and improved selection properties.

The remainder of this article is structured as follows. In Section 2, the underlying joint model as well as the `JMalct` boosting algorithm are formulated. Section 3 then applies the proposed method to simulated data with varying amounts of candidate variables. Several real-world applications are presented in Section 4 and the final section gives a brief summary and outlook.

## 2. Methods

This section first formulates the considered joint model as well as the basics of the underlying `JMboost` approach. Afterwards, the new `JMalct` routine and a thorough discussion of its computational details are provided.

### 2.1. Model Specification

A joint model consists of two sub-models modeling the longitudinal and time-to-event outcome, respectively. The longitudinal sub-model is specified as a linear mixed model

$$
\begin{aligned}
y_{ij} &= \eta_{\text{long}}(t_{ij}, \boldsymbol{x}_{\text{long}i}) + \varepsilon_{ij} \\
&= \beta_0 + \beta_t t_{ij} + \boldsymbol{\beta}_{\text{long}}^T \boldsymbol{x}_{\text{long}i} + \gamma_{0i} + \gamma_{ti} t_{ij} + \varepsilon_{ij},
\end{aligned}
\tag{1}
$$

with individuals $i = 1, \ldots, n$ and corresponding measurements $j = 1, \ldots, n_i$. Here, $\boldsymbol{x}_{\text{long}i} \in \mathbb{R}^{p_{\text{long}}}$ denotes a set of longitudinal time-independent covariates, and $t_{ij}$ the specific measurement times and normal distributed error components, i.e., $(\gamma_{0i}, \gamma_{ti}) \sim \mathcal{N}^{\otimes 2}(\boldsymbol{0}, \boldsymbol{Q})$ and $\varepsilon_{ij} \sim \mathcal{N}(0, \sigma^2)$ are assumed.

In the survival sub-model, the individual hazard is modeled by

$$
\lambda_i(t) = \lambda_0(t) \exp\left( \eta_{\text{surv}}(\boldsymbol{x}_{\text{surv}i}) + \alpha \eta_{\text{long}}(t, \boldsymbol{x}_{\text{long}i}) \right)
\tag{2}
$$

with the survival predictor $\eta_{\text{surv}}(\boldsymbol{x}_{\text{surv}i}) = \boldsymbol{\beta}_{\text{surv}}^T \boldsymbol{x}_{\text{surv}i}$ including baseline covariates $\boldsymbol{x}_{\text{surv}i} \in \mathbb{R}^{p_{\text{surv}}}$ and the longitudinal predictor $\eta_{\text{long}}$ reappearing in the survival sub-model, this time scaled by the association parameter $\alpha$. The baseline hazard $\lambda_0(t) := \lambda_0 > 0$ is chosen to be constant as conventional gradient boosting methods tend to struggle with a proper estimation of time-varying baseline hazard functions [26].

Given the sub-models (1) and (2) and assuming independence between the random components, the joint log-likelihood is

$$
\begin{aligned}
\ell(\eta_{\text{long}}, \eta_{\text{surv}}, \alpha, \lambda_0, \sigma^2 | \boldsymbol{y}, \boldsymbol{T}, \boldsymbol{\delta}) = \sum_{i=1}^{n} \Bigg\{ & \sum_{j=1}^{n_i} \log \phi\Big(y_{ij} | \eta_{\text{long}}(t_{ij}, \boldsymbol{x}_{\text{long}i}), \sigma^2\Big) \\
& + \delta_i \log \lambda_i(T_i | \eta_{\text{long}}, \eta_{\text{surv}}, \alpha, \lambda_0) - \int_0^{T_i} \exp(\lambda_i(t | \eta_{\text{long}}, \eta_{\text{surv}}, \alpha, \lambda_0)) dt \Bigg\},
\end{aligned}
\tag{3}
$$

where, in the longitudinal part, $\phi(\cdot | m, v)$ denotes the density of a normal distribution with mean $m$ and variance $v$. In this context, we considered the complete data log-likelihood as it is used solely for allocation purposes. The random effects will be estimated in a less time-consuming way based on a fixed penalization integrated in the random effects base-learner discussed in Section 2.4.

### 2.2. The `JMboost` Concept

In [12], joint models were estimated for the first time using a boosting algorithm, although they addressed a slightly different model to the one described above. The original concept in this publication was based on an alternating technique that used two loops: one outer loop circling through the two sub-predictors and two inner loops that circle through the single base-learners. In a very simple manner, the boosting algorithm can hence be summarized as follows:

- Initialize $\eta_{\text{long}}, \eta_{\text{surv}}, \alpha, \lambda_0$ and $\sigma^2$;
- While $m \leq \max(m_{\text{stop,l}}, m_{\text{stop,s}})$;
  - If $m \leq m_{\text{stop,l}}$: perform one boosting cycle to update $\eta_{\text{long}}$;
  - If $m \leq m_{\text{stop,s}}$: perform one boosting cycle to update $\eta_{\text{surv}}$;
  - If $m \leq m_{\text{stop,l}}$: update $\sigma^2$;
  - If $m \leq m_{\text{stop,s}}$: update $\lambda_0$ and $\alpha$.

Both sub-predictors have their own stopping iteration $m_{\text{stop,l}}$ and $m_{\text{stop,s}}$, which need to be optimized via a grid search. The latter is computationally quite burdensome, particularly for high numbers of candidate variables.

### 2.3. The `JMalct` Boosting Algorithm

The central `JMalct` algorithm is depicted in Algorithm 1.

---

**Algorithm 1:** JMalct

---

- **Initialize** predictors $\hat{\eta}_{\text{long}}^{[0]}$ and $\hat{\eta}_{\text{surv}}^{[0]}$. Specify base-learners $h_{\text{long}1}, \ldots, h_{\text{long}p}$ and $h_{\text{surv}1}, \ldots, h_{\text{surv}p}$, as well as $h_{\gamma}$. Initialize association $\hat{\alpha}^{[0]}$ and baseline hazard $\hat{\lambda}_0^{[0]}$. Choose iteration limit $m_{\text{stop}}$ and learning rate $c$, and define the sets $\mathcal{S}_{\text{long}}^{[0]} = \mathcal{S}_{\text{surv}}^{[0]} := \{1, \ldots, p\}$.
- **for** $m = 1$ to $m_{\text{stop}}$ **do**
  **step1: Allocation step**

Compute the gradients

$$\boldsymbol{u}_{\text{long}}^{[m]} = \left(u_{\text{long}ij}^{[m]}\right)_{i \leq n, j \leq n_i} = \left(y_{ij} - \hat{\eta}_{\text{long}ij}^{[m-1]}\right)_{i \leq n, j \leq n_i} \tag{4}$$

and

$$\boldsymbol{u}_{\text{surv}}^{[m]} = \left(u_{\text{surv}i}^{[m]}\right)_{i \leq n} = \left(\delta_i - \int_0^{T_i} \hat{\lambda}_i^{[m-1]}(t, \cdot) dt\right)_{i \leq n}. \tag{5}$$

Fit both gradients separately to the base-learners

$$\boldsymbol{u}_{\text{long}}^{[m]} \xrightarrow{\text{base-learner}} \hat{h}_{\text{long}r}^{[m]}, \quad r \in \mathcal{S}_{\text{long}}^{[m-1]}, \tag{6}$$

$$\boldsymbol{u}_{\text{surv}}^{[m]} \xrightarrow{\text{base-learner}} \hat{h}_{\text{surv}r}^{[m]}, \quad r \in \mathcal{S}_{\text{surv}}^{[m-1]}, \tag{7}$$

and select the best performing component for each predictor:

$$r_{\text{long}}^* = \arg\min_{r \leq p} \sum_{ij} \left(u_{\text{long}ij}^{[m]} - \hat{h}_{\text{long}ij}^{[m]}\right)^2, \quad r_{\text{surv}}^* = \arg\min_{r \leq p} \sum_i \left(u_{\text{surv}i}^{[m]} - \hat{h}_{\text{surv}i}^{[m]}\right)^2 \tag{8}$$

Compute the optimal step lengths $\nu_{\text{long}r^*}, \nu_{\text{surv}r^*}$ with corresponding likelihood values $\ell_{\text{long}r^*}, \ell_{\text{surv}r^*}$ and only update the component resulting in the best joint likelihood improvement:

$$\hat{\eta}_{\text{long}}^{[m]} = \hat{\eta}_{\text{long}}^{[m-1]} + c\nu_{\text{long}r^*}\hat{h}_{\text{long}r^*}^{[m]}, \quad \text{if } \ell_{\text{long}r^*} > \ell_{\text{surv}r^*}, \tag{9}$$

$$\hat{\eta}_{\text{surv}}^{[m]} = \hat{\eta}_{\text{surv}}^{[m-1]} + c\nu_{\text{surv}r^*}\hat{h}_{\text{surv}r^*}^{[m]}, \quad \text{if } \ell_{\text{long}r^*} < \ell_{\text{surv}r^*} \tag{10}$$

Update the active sets

$$\mathcal{S}_{\text{long}}^{[m]} = \mathcal{S}_{\text{long}}^{[m-1]} \setminus \{r_{\text{surv}}^*\}, \quad \text{if } \ell_{\text{surv}r^*} > \ell_{\text{long}r^*}, \tag{11}$$

$$\mathcal{S}_{\text{surv}}^{[m]} = \mathcal{S}_{\text{surv}}^{[m-1]} \setminus \{r_{\text{long}}^*\}, \quad \text{if } \ell_{\text{long}r^*} > \ell_{\text{surv}r^*}. \tag{12}$$

  **step2: Update remaining parameters**
Perform an additional longitudinal boosting update regarding the random structure:

$$\boldsymbol{u}_{\text{long}}^{[m]} \xrightarrow{\text{base-learner}} \hat{h}_{\gamma}^{[m]} \quad \Rightarrow \quad \hat{\eta}_{\text{long}}^{[m]} = \hat{\eta}_{\text{long}}^{[m]} + c\hat{h}_{\gamma}^{[m]} \tag{13}$$

Obtain updates for the association by maximizing the joint likelihood:

$$\hat{\alpha}^{[m]} = \arg\max_{\alpha \in \mathbb{R}} \ell(\alpha, \cdot) \tag{14}$$

  **end for**
- **Stop** the algorithm early based on probing, i.e., when a phantom variable would get selected.

---

### 2.4. Computational Details of the JMalct Algorithm

In the new JMalct algorithm, we only have one cycle. This cycle consists of three steps: in the first one, the base-learner with the best fitting gradient for the longitudinal predictor

$\eta_{\text{long}}$ is chosen and the corresponding step length $\nu_{\text{long}}$ is calculated. In the second step, the base-learner with the best fitting gradient for the time-to-event submodel $\eta_{\text{surv}}$ is chosen and the corresponding step length $\nu_{\text{surv}}$ is calculated. These first two steps will be referred to as the *G-steps* (gradient-steps) in the following. In the third step, referred to as the *L-step* (likelihood step), the likelihood is calculated for both the best longitudinal base-learner, weighted with the step length $\nu_{\text{long}}$, and the best survival base-learner, weighted with the step length $\nu_{\text{surv}}$. The base-learner performing better in the L-step is then chosen to be updated. The algorithm is summarized in the following overview and depicted in Figure 1. A detailed description is provided below.

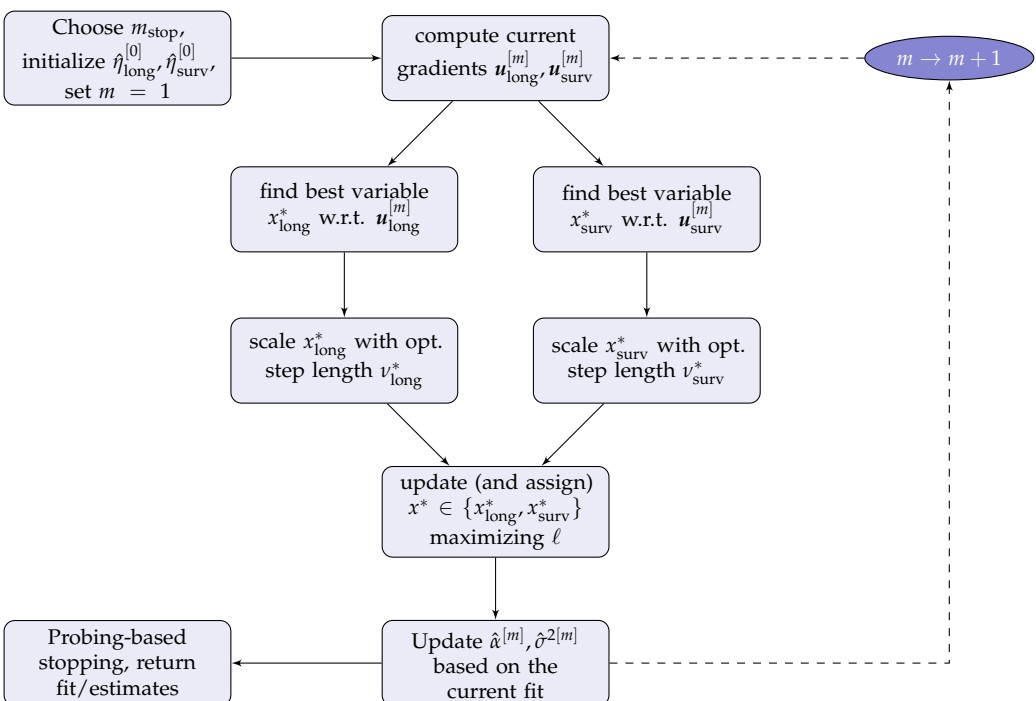

**Figure 1.** Schematic overview of the JMalct procedure.

`while` $m \leq m_{\text{stop}}$:

- *G-step 1*

  – Fit all base-learners to the longitudinal gradient with regard to $\hat{\eta}_{\text{long}}^{[m]}$;
  – Find the best-performer, $\beta_{\text{long}}^*$ and corresponding step-length $\nu_{long}$.

- *G-step 2*

  – Fit all base-learners to gradient with regard to $\hat{\eta}_{\text{surv}}^{[m]}$;
  – Find the best-performer, $\beta_{\text{surv}}^*$ and corresponding step-length $\nu_{surv}$.

- *L-step*

  – Fit likelihood for $\eta_{\text{long}}^*$ and $\eta_{\text{surv}}^*$ with updates from G1 and G2;
  – Select the best-performer and update corresponding sub-predictor;
  – Remove the selected candidate variable from options to choose for the other predictor (if not performed already).

- *Step 4*

  – Update $\hat{\alpha}^{[m]}, \hat{\sigma}^{2[m]}$ based on the current fit.

The baseline covariates that enter the allocation process are not assigned to a sub-model in the beginning and therefore have to be considered in two forms. $X_{\text{long}} \in \text{Mat}_{\mathbb{R}}(N, p)$, where $N = \sum_i n_i$, denotes the set of candidate variables resembled as longitudinal covariates, i.e., measurements assigned to the same individual $i$ contain the same cluster-constant

measurement $n_i$ times. On the other hand, $X_{\text{surv}} \in \text{Mat}_{\mathbb{R}}(n, p)$ contains the exact same variables as $X_{\text{long}}$ but reduced to just one representative of each individual in order to fit the corresponding base-learner to the survival gradient. The measurements of one specific covariate $r$ are denoted by $x_{\text{long}r}$ and $x_{\text{surv}r}$, which matches the $r$th column of the corresponding matrix.

**Starting values.** The regression coefficients underlying the allocation mechanism are necessarily set to zero, i.e., $\boldsymbol{\beta}_{\text{long}}^{[0]} = \boldsymbol{\beta}_{\text{surv}}^{[0]} = \mathbf{0}$. The remaining longitudinal parameters are extracted by an initial linear mixed model fit

$$y = \beta_0 + \beta_t \cdot t + \gamma_0 + \gamma_t \cdot t \tag{15}$$

containing only the intercept as well as time and random effects. For the remaining survival parameters, we chose $\alpha^{[0]} = 0$ and $\hat{\lambda}^{[0]} = \sum_i \delta_i / \sum_i T_i$.

**Computing the gradients.** The gradients $u_{\text{long}}$ and $u_{\text{surv}}$ are a crucial component of the `JMalct` algorithm. For the longitudinal part, we considered the quadratic loss $\rho(y, \eta) = \frac{1}{2}(y - \eta)^2$ and calculated

$$u_{\text{long}} = -\frac{\partial \rho}{\partial \eta_{\text{long}}}(y, \eta_{\text{long}}) = y - \eta_{\text{long}} \tag{16}$$

as the regular residuals of the longitudinal sub-model, following [9]. The survival gradient was obtained by differentiating the likelihood (3) with respect to $\eta_{\text{surv}}$, yielding

$$u_{\text{surv}} = -\frac{\partial \rho}{\partial \eta_{\text{surv}}}(\eta_{\text{surv}}, \cdot) = \left(\delta_i - \int_0^{T_i} \hat{\lambda}_i^{[m-1]}(t, \cdot)dt\right)_{i \leq n}, \tag{17}$$

as the longitudinal part vanishes. This is a nice analogy to the longitudinal gradient, as $u_{\text{surv}}$ represents the martingale residuals of the survival sub-model.

**Fitting the longitudinal base-learners.** The possible fixed effects estimates were obtained by fitting the pre-specified base-learners to the longitudinal and survival gradient. In the longitudinal case, the fixed effects base-learners $h_{\text{long}1}, \ldots, h_{\text{long}p}$ were equipped with an additional effect estimate for the time coefficient $\beta_t$ as this variable shall not be subject to the selection and allocation mechanism. Fitting the base-learners is achieved by

$$\hat{h}_{\text{long}r} = S_{\text{long}r} u_{\text{long}}, \quad r = 1, \ldots, p, \tag{18}$$

with the projection matrix

$$S_{\text{long}r} = \tilde{x}_{\text{long}r}(\tilde{x}_{\text{long}r}^T \tilde{x}_{\text{long}r})^{-1}\tilde{x}_{\text{long}r}^T, \quad r = 1, \ldots, p, \tag{19}$$

where $\tilde{x}_{\text{long}r} = (\mathbf{1}, t, x_{\text{long}r})$ and $t$ denotes the collection of longitudinal measurement times. If the base-learner actually gets selected, estimates $\hat{\beta}_0$ for the intercept and $\hat{\beta}_t$ for the time effect receive the corresponding updates computed in the fitting process.

**Fitting the survival base-learners.** Similar to the longitudinal part, the survival base-learner was fitted by applying the corresponding projection matrix to the survival gradient, i.e.,

$$\hat{h}_{\text{surv}r} = S_{\text{surv}r} u_{\text{surv}}, \quad r = 1, \ldots, p, \tag{20}$$

where the survival gradient $u_{\text{surv}}$ represents the martingale residuals of the time-to-event model. The projection matrix takes the form

$$S_{\text{surv}r} = \tilde{x}_{\text{surv}r}(\tilde{x}_{\text{surv}r}^T \tilde{x}_{\text{surv}r})^{-1}\tilde{x}_{\text{surv}r}^T, \quad r = 1, \ldots, p, \tag{21}$$

with $\tilde{x}_{\text{surv}r} = (\mathbf{1}, x_{\text{surv}r})$. This means that, if the base-learner actually gets selected, the estimate $\hat{\lambda}_0$ for the constant baseline hazard receives the corresponding update computed in the fitting process.

**Adaptive step lengths.** As the two distinct sub-models affect different parts of the joint likelihood, it may not be sufficient to stick to a fixed learning rate, e.g., $\nu_{\text{long}} = \nu_{\text{surv}} = 0.1$. To ensure that the comparison of potential likelihood improvements is fair, for each selected component, the optimal step length was computed using a basic line search by finding

$$\nu_{\text{long}} = \underset{\nu \in \mathbb{R}^+}{\arg\max} \, \ell(\hat{\eta}_{\text{long}} + \nu \hat{h}_{\text{long}r^*_{\text{long}}}, \cdot), \quad \nu_{\text{surv}} = \underset{\nu \in \mathbb{R}^+}{\arg\max} \, \ell(\hat{\eta}_{\text{surv}} + \nu \hat{h}_{\text{surv}r^*_{\text{surv}}}, \cdot). \quad (22)$$

following [23]. The corresponding maximal likelihood values are denoted by $\ell^*_{\text{long}}$ and $\ell^*_{\text{surv}}$, which were used to determine the overall best-performing sub-model of each iteration. When this is achieved, the learning rate for the actual update was then again scaled by a constant $c < 1$—here, $c = 0.1$—in order to ensure small updates with weak base-learners.

**Fitting the random effects base-learner.** In general, the random effects base-learner is similar to the formulation found in the appendix of [27]. One major difference is that it is fixed for all iterations and not updated based on the current covariance structure. It is defined through its projection matrix

$$S_\gamma = \mathbf{Z}C(\mathbf{Z}^T\mathbf{Z} + \lambda_{\text{df}})^{-1}\mathbf{Z}^T \quad (23)$$

where $\lambda_{\text{df}}$ was chosen so that $\text{tr}(S_\gamma) = \text{df}$ holds, which fixes the degrees of freedom for the random effects update. In the simulation study, we used $\text{df} = 10$ and determined the corresponding $\lambda_{\text{df}}$ with the internal function `mboost:::df2lambda()`.

The matrix $\mathbf{Z}$ denotes the conventional random effects design matrix for intercepts and slopes, i.e.,

$$\mathbf{Z} = \text{diag}(\mathbf{Z}_1, \ldots, \mathbf{Z}_n), \quad \mathbf{Z}_i = \begin{pmatrix} 1 & t_{i1} \\ \vdots & \vdots \\ 1 & t_{in_i} \end{pmatrix}, \quad i = 1, \ldots, n, \quad (24)$$

and $C$ is a correction matrix introduced in [28] correcting the random effects update for the candidate variables $x_{\text{long}1}, \ldots, x_{\text{long}p}$, which are baseline covariates and thus cluster-constant. A derivation of the correction matrix $C$ can also be found in Appendix A.

**Tuning the hyperparameter $m$ based on probing.** Both the step length as well as the number of iterations can be considered as hyperparameters of the boosting algorithm. Since the step length is usually set as constant or, like in this work, determined by an adaptive line search, the number of overall iterations $m$ states the main tuning parameter of the algorithm. While this hyperparameter is usually tuned in a computationally more extensive way by considering out-of-bag loss, we determined the optimal amount $m^*$ with the help of probing. Probing for gradient boosting was introduced by [25]. The pragmatic idea avoids more time-consuming procedures such as cross validation or bootstrapping, which rely on a re-fitting of the model. For each covariate $x_r$, another variable $\bar{x}_r$ was added to the set of candidate variables, where $\bar{x}_r$ is a random permutation of the observations contained in $x_r$. These additional variables were artificially created to be non-informative and called *probes* or *shadow variables*. Instead of finding the best-performing number of iterations based on a computationally burdensome cross validation, the boosting routine was simply stopped as soon as one of the shadow variables $\bar{x}_r$, i.e., a known-to-be non-informative variable, would get selected. The focus is hence shifted from tuning the algorithm purely based on prediction accuracy (with regard to the test risk) towards a reasonable variable selection.

**Computational complexity and asymptotic behavior.** Due to the artificial construction of the algorithm and the comparatively complex model class, theoretical analysis regarding complexity and asymptotic behavior is quite a challenging task. The model-based boosting related literature is still little-developed with respect to theoretical investigations, but thorough analyses in simpler cases were carried out in [29] for the quadratic loss, where exponentially fast bias reduction could be proven, as well as for more general settings in

[30,31]. Consistency properties for very-high-dimensional linear models were obtained in [32] and, regarding JMalct, we refer to the following section, where further insights with respect to the algorithm's complexity are given based on numerical evaluations. In addition, we experienced no convergence issues in simulations and applications.

## 3. Simulation Study

The JMalct algorithm was evaluated by conducting a simulation study where data according to the assumed generating process specified in Section 2.1 were simulated and models were subsequently fitted using JMalct and, if sensible, JM [3] as a benchmark and well-established approach. In addition, we considered the combination JMalct+JM, where JMalct was used solely for allocating the variables, which were then refitted by JM according to the allocation obtained from JMalct. After briefly highlighting the single scenarios, the simulation section evaluates allocation properties and the accuracy of estimates, as well as the quality of the prediction and the computational burden.

### 3.1. Setup

We simulated data according to the model specification in Section 2.1 with $n = 500$ and $n_i = 5$ using inversion sampling. The pre-specified true parameter values are

$$\beta_0 = 1, \quad \beta_t = 1.5, \quad \beta_{\text{long}}^T = (1, 2, 1, 2), \quad \beta_{\text{surv}}^T = (0.3, 0.5, 0.3, 0.5), \quad \alpha = 0.1 \quad (25)$$

with variance components

$$\sigma = 0.1, \quad Q = \begin{pmatrix} \tau_0^2 & 0 \\ 0 & \tau_t^2 \end{pmatrix}, \quad \tau_0 = 2, \quad \tau_t = 0.3. \quad (26)$$

The entries of the covariate vectors $x_{\text{long}i}$ and $x_{\text{surv}i}$ were drawn independently from the uniform distribution $\mathcal{U}([-0.1, 0.1])$. In addition to the informative covariates with effects $\beta_{\text{long}}$ and $\beta_{\text{surv}}$, the total set of covariates was expanded with a varying number $p_{\text{non-inf}}$ of non-informative noise variables. The baseline hazard was chosen as $\lambda_0(t) \equiv 1$ and given the censoring mechanism described in Algorithm A1 depicted in Appendix B. The chosen parameter values result in an average censoring rate of $\approx 50\%$. All of the parameters were specified in a way to obtain reasonably distributed event times $T$.

Overall, we considered four scenarios with varying numbers of additional noise variables $p_{\text{non-inf}}$ yielding overall dimensions $P \in \{10, 25, 50, 100\}$. In each scenario, 100 independent data sets were generated and models were fitted using the various routines. The results were then summarized over all 100 independent simulation runs.

### 3.2. Selection and Allocation

In order to address allocation, we considered the criteria of correctly allocated (CA) and incorrectly allocated (IA) variables per predictor, as well as the share of false positives (FPs). Precisely, $\text{CA}_{\text{long}}$ is the share of longitudinal variables, which are correctly assigned to the longitudinal predictor, and $\text{IA}_{\text{long}}$ is the share of survival variables, which are falsely assigned to the longitudinal predictor and $\text{CA}_{\text{surv}}$, $\text{IA}_{\text{surv}}$ analogously. FPs, on the other hand, denote the share of wrongly selected noise variables regardless of which predictor they are assigned to.

Table 1 depicts allocation and selection properties obtained for the different simulation scenarios. While, for the longitudinal predictor, variables get allocated perfectly, the survival part shows less ideal but still satisfactory results. There are various possible explanations for this behavior. On the one hand, the simulated signal is less strong for the survival effects due to the chosen parameter values, which, in general, increases the chance of false negatives. On the other hand, the longitudinal part of the likelihood carries more information, as there are more longitudinal measurements available than event times, which increases the risk of incorrect allocations. Finally, survival variables being incorrectly allocated to the longitudinal predictor is inherently more probable than vice versa as the

longitudinal predictor also appears in the survival sub-model and the model therefore still accounts for the variables' impact on the time-to-event outcome. While the allocation properties are roughly constant with a varying number of dimensions, the false positives rate clearly diminishes with more and more noise variables.

**Table 1.** Share of correctly allocated (CA) and incorrectly allocated (IA) variables for each predictor as well, as false positive rate. Values are averaged over 100 independent simulation runs of each scenario.

| P | CA$_\text{long}$ | IA$_\text{long}$ | CA$_\text{surv}$ | IA$_\text{surv}$ | FP |
|---|---|---|---|---|---|
| 10 | 1.00 | 0.20 | 0.76 | 0.00 | 0.58 |
| 25 | 1.00 | 0.15 | 0.78 | 0.00 | 0.12 |
| 50 | 1.00 | 0.14 | 0.77 | 0.00 | 0.05 |
| 100 | 1.00 | 0.10 | 0.76 | 0.00 | 0.02 |

### 3.3. Estimation Accuracy

The accuracy of coefficient estimation is shown in Table 2, separated for each sub-model. We considered the mean squared error (mse) computed as

$$\text{mse}_\text{long} = \|\theta_\text{long} - \hat{\theta}_\text{long}\|^2, \quad \text{mse}_\text{surv} = \|\theta_\text{surv} - \hat{\theta}_\text{surv}\|^2, \tag{27}$$

where $\theta_\text{long} = (\beta_0, \beta_t, \boldsymbol{\beta}_\text{long}^T)^T$ and $\theta_\text{surv} = (\lambda, \alpha, \boldsymbol{\beta}_\text{surv}^T)^T$. The lower half of the table discards all entries of the estimates $\hat{\boldsymbol{\beta}}_\text{long}$ and $\hat{\boldsymbol{\beta}}_\text{surv}$ referring to non-informative variables and thus only measures the accuracy of the effects that are known to be informative.

It is evident that the accuracy of JM is heavily influenced by the number of noise variables, whereas the routines relying on the allocation and selection mechanism by JMalct stay fairly robust. As usual for regularization techniques, JMalct's estimates for informative effects are slightly biased due to the early stopping of the algorithm. The combination JMalct+JM, however, stays unaffected by the number of noise variables and is, at least for the longitudinal predictor, the most accurate. The main hindrance of this approach is that the estimation accuracy of survival effects is slightly more influenced by false negatives occurring in the selection mechanism, which is why the combination lags behind its two competitors regarding precision for the survival sub-model.

**Table 2.** Mean squared error for longitudinal (mse$_\text{long}$) and survival (mse$_\text{surv}$) coefficients averaged over 100 independent simulation runs. Regular parameter estimates are indicated by $\theta$, whereas $\theta_{-\text{n.inf}}$ denotes the second half, where non-informative effects are neglected.

| | P | JMalct | | JM | | JMalct+JM | |
|---|---|---|---|---|---|---|---|
| | | mse$_\text{long}$ | mse$_\text{surv}$ | mse$_\text{long}$ | mse$_\text{surv}$ | mse$_\text{long}$ | mse$_\text{surv}$ |
| $\theta$ | 10 | 0.497 | 0.343 | 0.713 | 0.489 | 0.342 | 0.453 |
| | 25 | 0.479 | 0.296 | 1.888 | 1.359 | 0.309 | 0.417 |
| | 50 | 0.585 | 0.303 | 4.212 | 3.375 | 0.374 | 0.409 |
| | 100 | 0.563 | 0.295 | 9.532 | 9.483 | 0.320 | 0.393 |
| $\theta_{-\text{n.inf}}$ | 10 | 0.486 | 0.323 | 0.302 | 0.254 | 0.288 | 0.423 |
| | 25 | 0.470 | 0.292 | 0.247 | 0.291 | 0.239 | 0.398 |
| | 50 | 0.578 | 0.297 | 0.362 | 0.367 | 0.306 | 0.384 |
| | 100 | 0.557 | 0.290 | 0.315 | 0.558 | 0.254 | 0.373 |

### 3.4. Predictive Performance

Boosting is a tool primarily designed for prediction, and thus the predictive performance of JMalct and how it compares to established routines are of interest. Since our underlying joint model focuses on the time-to-event outcome as the main endpoint, we

evaluated the prediction accuracy regarding the predicted and actual event time based on additional test data with $n_{\text{test}} = 1000$ individuals and $n_i = 5$. We considered the loss

$$L(T, \hat{T}) = |\log T - \log \hat{T}|, \quad \hat{T} = \mathbb{E}[T], \tag{28}$$

as the absolute deviation between the predicted and actual event time $\hat{T}$ and $T$, respectively, on a log-scale [33].

Figure 2 depicts the values of $L$ over the varying numbers of additional noise variables. The prediction is comparable among the three routines in low-dimensional settings. However, as expected, it worsens for `JM` when the dimensions increase. Both `JMalct` and the combination `JMalct+JM` rely on the selection conducted by `JMalct` and hence produce sparse models, which is why their quality of prediction stays fairly equal even in higher dimensions.

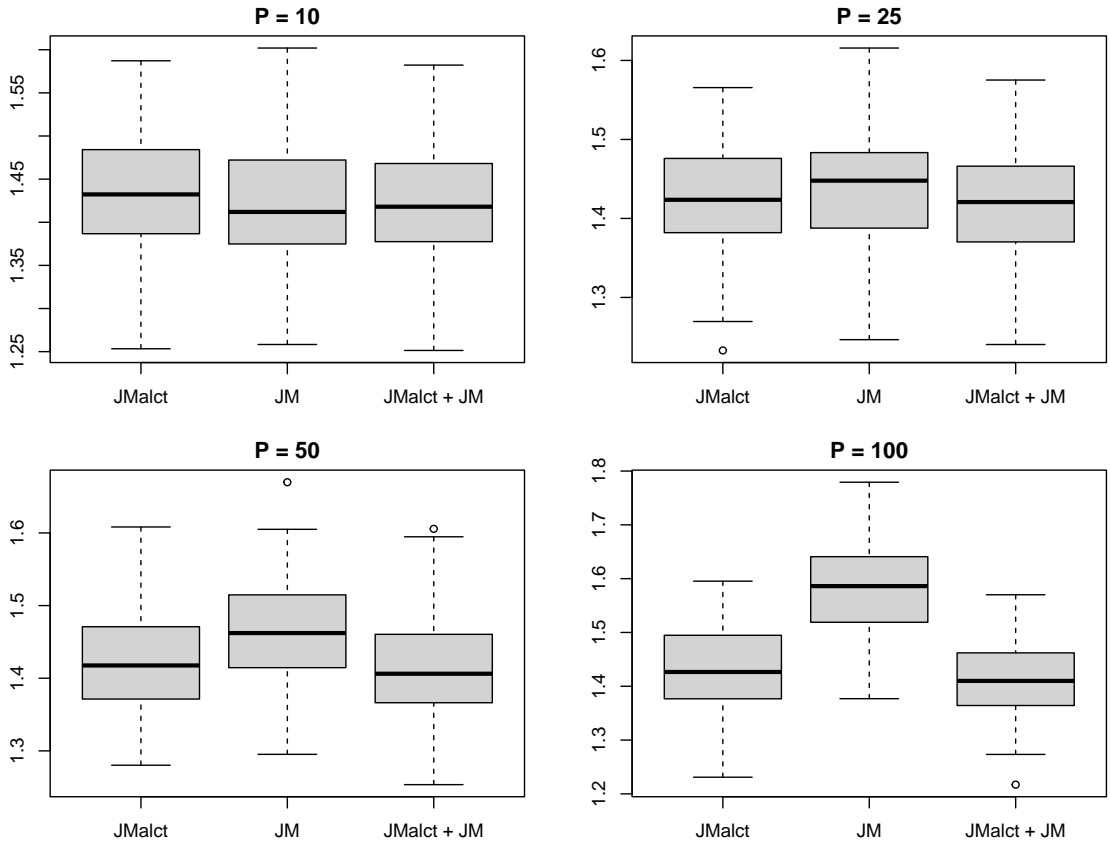

**Figure 2.** Comparison of the prediction error ($L$) of the survival part for the varying numbers of non-informative noise variables.

### 3.5. Computational Effort

Table 3 shows the elapsed computation time measured in seconds, where each simulation run was carried out on a *2 x 2.66 GHz-6-Core Intel Xeon* CPU (*64GB* RAM). Most obviously, `JM` becomes tremendously more burdensome as the dimensions increase. The constant or even a little decreasing computation times for `JMalct` over various dimensions might be surprising at first, as component-wise procedures such as gradient boosting tend to increase at least linearly in computation time with additional covariates. However, as the overall stopping criterion is based on probing, the algorithm tends to stop earlier in high-dimensional settings since more non-informative probes are available, increasing the probability that one might get selected earlier in the process. Due to the sparsity obtained by `JMalct`, the combination `JMalct+JM` also profits from the allocation and selection mechanism regarding the computational effort, as `JM` runs considerably faster again.

**Table 3.** Average computation times of the three approaches measured in seconds.

| P | JMalct | JM | JMalct+JM |
|---|---|---|---|
| 10 | 92.69 | 97.44 | 149.19 |
| 25 | 88.80 | 339.41 | 151.65 |
| 50 | 86.92 | 1009.52 | 152.59 |
| 100 | 85.06 | 3517.76 | 146.26 |

*3.6. Complexity*

While a formulation of explicit complexity results for the `JMalct` routine is quite technical in general, like that stated in Section 2.4, simulations can give insights toward how the algorithm scales up with varying numbers of observations and covariates. Therefore, we considered the same setup as above with different values for $n$ and $p$ and ran the `JMalct` routine 100 times independently for $m_{stop} = 100$ iterations without early stopping. Figure 3 depicts the averaged computation times for increasing values of $n$ and $p$.

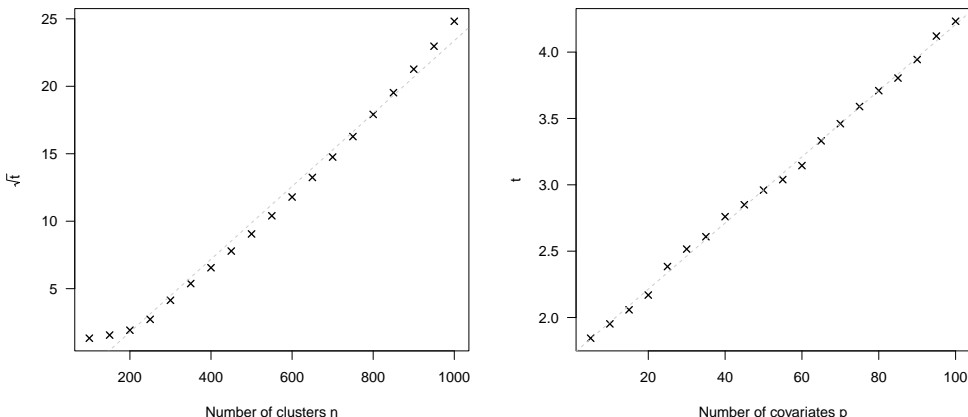

**Figure 3.** Average `JMalct` run times for varying numbers of clusters $n$ and covariates $p$. Dashed gray lines depict the corresponding linear model fit. Left panel shows square root times to highlight the quadratic relationship.

The left figure depicts square root computation times with $p = 3$ as fixed, and thus reveals quadratically growing run times for increasing observations. On the other hand, the run times clearly expose a linear relationship with the amount of total covariates and $n = 100$ as fixed. This is to be expected, as further candidate variables simply add to the inner loops of univariate base-learner fits and, thus, the algorithm is capable of fitting data sets with almost arbitrary high dimensions.

**4. 1994 AIDS Study**

The 1994 AIDS data [34] were originally collected in order to compare two antiretroviral drugs based on a collective of HIV-positive patients. They include 1405 longitudinal observations of 467 individuals, from which, 188 unfortunately died during the course of the study. The main longitudinal outcome is each patient's repeatedly measured CD4 cell counts. CD4 cells decline in HIV-positive patients and are a well-known proxy for disease progression, and are therefore of high interest. Apart from the CD4 cell count as the longitudinal outcome, death as the time-to-event outcome and time $t$ itself, the four additional baseline variables—drug (treatment group), `gender`, `AZT` (indicator of whether a previous AZT therapy failed) and `AIDS` (indicator of whether AIDS is diagnosed)—were observed. The structure of the data is depicted in Table 4.

**Table 4.** Structure of the data with primary outcomes for the joint analysis in the three columns on the left.

| $y$ | $T$ | $\delta$ | $t$ | Drug | Gender | AZT | prevOI | id |
|------|-------|----|----|------|--------|------------|--------|----|
| 10.67 | 16.97 | 0 | 0 | ddC | male | intolerance | AIDS | 1 |
| 8.43 | 16.97 | 0 | 6 | ddC | male | intolerance | AIDS | 1 |
| 9.43 | 16.97 | 0 | 12 | ddC | male | intolerance | AIDS | 1 |
| 6.32 | 19.00 | 0 | 0 | ddI | male | intolerance | noAIDS | 2 |
| 8.12 | 19.00 | 0 | 6 | ddI | male | intolerance | noAIDS | 2 |
| 4.58 | 19.00 | 0 | 12 | ddI | male | intolerance | noAIDS | 2 |
| 5.00 | 19.00 | 0 | 18 | ddI | male | intolerance | noAIDS | 2 |
| 3.46 | 18.53 | 0 | 0 | ddI | female | intolerance | AIDS | 3 |
| 3.61 | 18.53 | 0 | 2 | ddI | female | intolerance | AIDS | 3 |
| 6.16 | 18.53 | 1 | 6 | ddI | female | intolerance | AIDS | 3 |
| ⋮ | ⋮ | ⋮ | ⋮ | ⋮ | ⋮ | ⋮ | ⋮ | ⋮ |

Figure 4 depicts the coefficient paths computed by the `JMalct` algorithm and the corresponding allocation process. The variable `AIDS` is selected into the longitudinal sub-model right away and frequently updated. This is not surprising, as the diagnosis of AIDS is by definition partly linked to the CD4 cell count. The remaining variables `drug` and `gender` were also allocated to the longitudinal sub-model by a smaller amount, whereas `AZT` was selected into the survival predictor, indicating an increased risk of death for patients with failed AZT therapy.

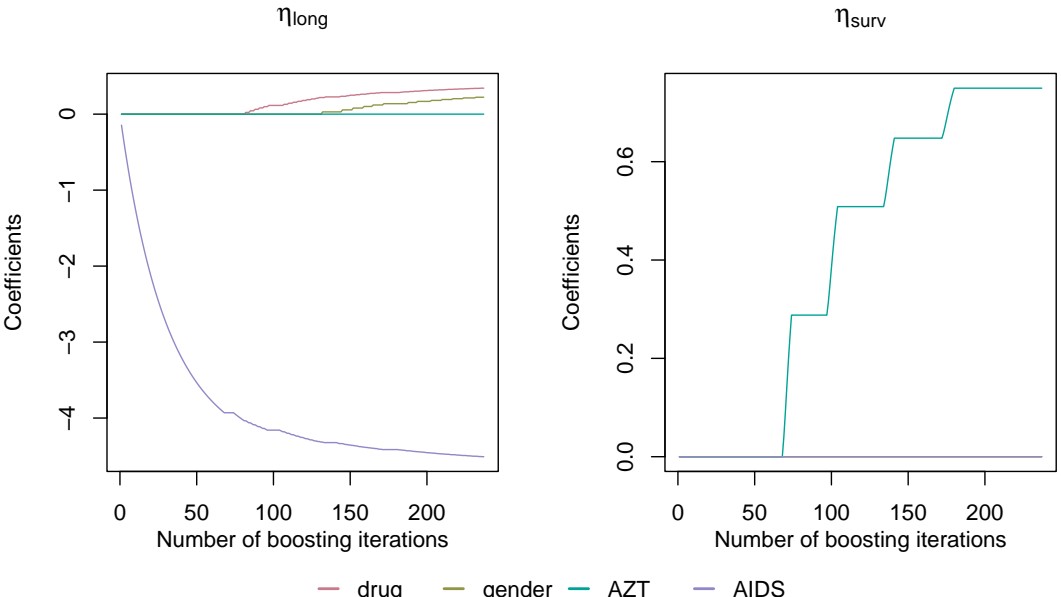

**Figure 4.** Coefficient progression in both sub-models for AIDS data. The variable `AZT` was assigned to $\eta_{\text{surv}}$, and the rest to $\eta_{\text{long}}$.

## 5. Discussion and Outlook

Finding adequate data-driven allocation mechanisms for joint models is a very important task, as modeling possibilities increase exponentially with a growing number of covariates. Until today, decisions about the specific model choice have to be made based on background knowledge or by conducting a preliminary analysis, and both of these approaches can be seen as rather unsatisfactory.

The `JMalct` algorithm combines recent findings from the field of gradient boosting to construct a fast-performing allocation and selection mechanism for a joint model focusing on time-to-event data as the primary outcome. A simulation study revealed that the

selection and allocation mechanism yields promising results while preserving the well-known advantages from gradient boosting. Therefore, it is advised to use the `JMalct` algorithm in its current form in advance of the actual analysis in order to determine an allocation of covariates, which is then fitted using convenient frameworks such as `JM`.

Possible ways of improving the accuracy of estimates and allocation properties regarding the survival sub-model could be based on additional weighting rules. As the longitudinal part contributes substantially more to the likelihood due the higher number of observations, weighting the two sub-models solely by different step lengths may not be sufficient. Promising ideas are initial weightings of the sub-models using various maximum likelihood estimations or focusing on a relative likelihood improvement in the selection step.

Another aspects focuses on variable selection and tuning the algorithm via probing. Although probing leads to fast runtimes and good selection properties, the procedure comes with disadvantages. Especially in higher dimensions, the probability that one shadow variable is informative simply by chance increases, leading to very early stopping. An alternative could rely on stability selection [35,36], as shown to be helpful in other cases [37].

Furthermore, the difference in the proportion of falsely selected variables between the longitudinal and survival sub-predictor could be an inherent joint modeling problem and should also be subject of future analysis. Further research is also warranted on theoretical insights, as it remains unclear if the existing findings on the consistency of boosting algorithms [32,38] also hold for the adapted versions for boosting joint models.

In conclusion, the `JMalct` algorithm represents a promising statistical inference scheme for joint models that also provides a starting point for a much wider framework of boosting joint models, covering a great range of potential models and types of predictor effects.

**Author Contributions:** Conceptualization, C.G. and E.B.; methodology, C.G., A.M. and E.B.; software, C.G. and E.B.; formal analysis, C.G.; investigation, C.G.; writing—original draft preparation, C.G.; writing—review and editing, E.B. and A.M.; project administration, E.B.; funding acquisition, E.B. All authors have read and agreed to the published version of the manuscript.

**Funding:** The work on this article was supported by the DFG (Number 426493614) and the Volkswagen Foundation (Freigeist Fellowship).

**Data Availability Statement:** All code and data required to reproduce the finding of this article are available.

**Acknowledgments:** The work on this article was supported by the DFG (Number 426493614) and the Volkswagen Foundation (Freigeist Fellowship). We further acknowledge the support by the Open Access Publication Funds of the University of Göttingen.

**Conflicts of Interest:** The authors declare no conflict of interest.

## Appendix A. Correction Matrix $C$

Due to the separated updating process for the random effects, it may be necessary to adjust the estimates for possible correlations with cluster-constant covariates using the correction matrix $C$. The following derivation is a special case of the more general version proposed in [28]. For the correction of the random intercepts $\tilde{\gamma}_0 = (\gamma_{01}, \ldots, \gamma_{0n})^T$ and random slopes $\tilde{\gamma}_t = (\gamma_{t1}, \ldots, \gamma_{tn})^T$ with the baseline covariates $X_{\text{surv}}$ defined in Section 2.4, consider the residual generating matrix

$$C_A = I_n - X_{\text{surv}}(X_{\text{surv}}^T X_{\text{surv}})^{-1} X_{\text{surv}} \tag{A1}$$

and subsequently $C_B = \text{diag}(C_A, C_A)$, so that the product $(C_B)\tilde{\gamma}$, $\tilde{\gamma} = (\tilde{\gamma}_0^T, \tilde{\gamma}_t^T)^T$ corrects the random intercepts $\tilde{\gamma}_0$ and slopes $\tilde{\gamma}_t$ for any covariates contained in the corresponding matrix $X_{\text{surv}}$ by counting out the orthogonal projections of the given random effect estimates on the subspace generated by the covariates $X_{\text{surv}}$. This ensures that the coefficient

estimate for the random effects is uncorrelated with any observed covariate. The final correction matrix $C$ is obtained by

$$C = P^{-1} C_B P, \tag{A2}$$

where $P$ is a permutation matrix mapping $\gamma = (\gamma_{01}, \gamma_{t1}, \dots, \gamma_{0n}, \gamma_{tn})$ to

$$P\gamma = \tilde{\gamma} \tag{A3}$$

and thus accounts for the usual ordering of the random effects in mixed-model frameworks.

### Appendix B. Simulation Algorithm

The following algorithm is used to generate data in Section 3.

---

**Algorithm A1:** `simJM`

---

- **Choose** integers $n, n_i$ and parameter values $\beta_0, \beta_t, \boldsymbol{\beta}_{\text{long}}, \boldsymbol{\beta}_{\text{surv}}$ and $\alpha$ with variance components $\sigma$ and $Q$. Specify a baseline hazard $\lambda_0(t)$.
- **Generate** $n \cdot n_i$ longitudinal measurement times mimicking *yearly appointments* the following way:
  - Sample $d_{ij} \sim \mathcal{U}(\{1, \dots, 365\})$ and set $\tilde{t}_{ij} := (j-1) \cdot 365 + d_{ij}$ for $i = 1, \dots, n$ and $j = 1, \dots, n_i$.
  - For each $i$, shift observation times to $\tilde{t}_{i1} = 0$.
  - Standardize time points to the unit interval by $t_{ij} := \tilde{t}_{ij} / (n_i \cdot 365)$.
- **Generate** covariate vectors $x_{\text{long}i}, x_{\text{surv}i}$ for $i = 1, \dots, n$ corresponding to the lengths of $\boldsymbol{\beta}_{\text{long}}$ and $\boldsymbol{\beta}_{\text{surv}}$.
- **Calculate** the longitudinal response

$$y_{ij} = \underbrace{\beta_0 + \beta_t t_{ij} + \boldsymbol{\beta}_{\text{long}}^T x_{\text{long}i} + \gamma_{0i} + \gamma_{ti} t_{ij}}_{\eta_{\text{long}}(t_{ij}, x_{\text{long}i})} + \varepsilon_{ij} \tag{A4}$$

with $\varepsilon_{ij} \sim \mathcal{N}(0, \sigma^2)$ and $(\gamma_{0i}, \gamma_{ti}) \sim \mathcal{N}^{\otimes 2}(\mathbf{0}, Q)$. Define hazard functions

$$\lambda_i(t) = \lambda_0(t) \exp\left( \boldsymbol{\beta}_{\text{surv}}^T x_{\text{surv}i} + \alpha \eta_{\text{long}}(t, x_{\text{long}i}) \right) \tag{A5}$$

as described in Section 2.1.

- **Draw** event times by generating random numbers $u_i \sim \mathcal{U}([0,1])$ and setting

$$T_i^* := F_i^{-1}(u), \quad F_i(t) = 1 - \exp\left( -\int_0^t \lambda_i(s) ds \right), \tag{A6}$$

according to inversion sampling.

- **Censor** by setting $T_i := \min(T_i^*, t_{in_i})$ to obtain censored data with censoring indicator $\delta_i := \mathbf{1}(T_i^* \le t_{in_i})$ and receive the *observed* survival outcome $(\boldsymbol{T}, \boldsymbol{\delta}) = (T_i, \delta_i)_{i=1,\dots,n}$.
- **Delete** all longitudinal observations corresponding to times $t_{ij} > T_i$ for every $i$.

---

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
