# Peer review of "Variable Selection and Allocation in Joint Models via Gradient Boosting Techniques"

_mathematics, doi:10.3390/math11020411_

Round 1

Reviewer 1 Report

The paper deals with an exciting topic and the results obtained by the authors sound great. However, the actual contribution of the paper needs to be better shown since the comparative study is not strengthened with enough comparisons with new approaches treating the same problem. Thus, the reviewer invites the authors to consider comparing their algorithm to more new state-of-the-art methods.

I have included below a few comments that I suggest for the attached paper.

1- The paper lacks the "related work" section. I suggest the authors perform an extensive study to review what people have done before.
2 - I suggest the authors add an illustrative example. It will clear up the idea of the paper.
3 - Adding the complexity of the proposed algorithm will surely be an asset.

Reviewer 2 Report

This paper introduces a technique that utilizes the gradient boosting for distributional regression to build a mechanisms to assign covariates to a signle sub-period in a joint manner. Their approach is the combination of non-cyclical updating with step-length adaptation. Here are a few points tha tcame to mind:

- what is the significance of this method over their countrparts specially Bayesain techniques? Bayesain techniques have shown very promising results in this context.

- How does the introduced method scale up?  and what is the computational costs? 

- How does the hyperparamters are learned? 

- Does this algorithm converge? if does what is the non-asymptotic behavior of it? what is the convergence rate?

- Regarding the simulations, I am uncertain how this method would work if the data was very high-dimesion. The real data is a bit old and not very high-dimension. What is the complexity of the algorithm for high-dimensional data? 

Round 2

Reviewer 2 Report

I'd like to thank the authors for the response and the revisions they made. Most of my concerns were addressed.